# Robust Manipulation Primitive Learning
# via Domain Contraction

**Teng Xue[1,2], Amirreza Razmjoo[1,2], Suhan Shetty[1,2], Sylvain Calinon[1, 2]**
[1]Idiap Research Institute    [2]École Polytechnique Fédérale de Lausanne (EPFL)

{teng.xue, amirreza.razmjoo, suhan.shetty, sylvain.calinon}@idiap.ch
https://sites.google.com/view/robustpl

**Policy training**       **Parameter-conditioned policy retrieval**

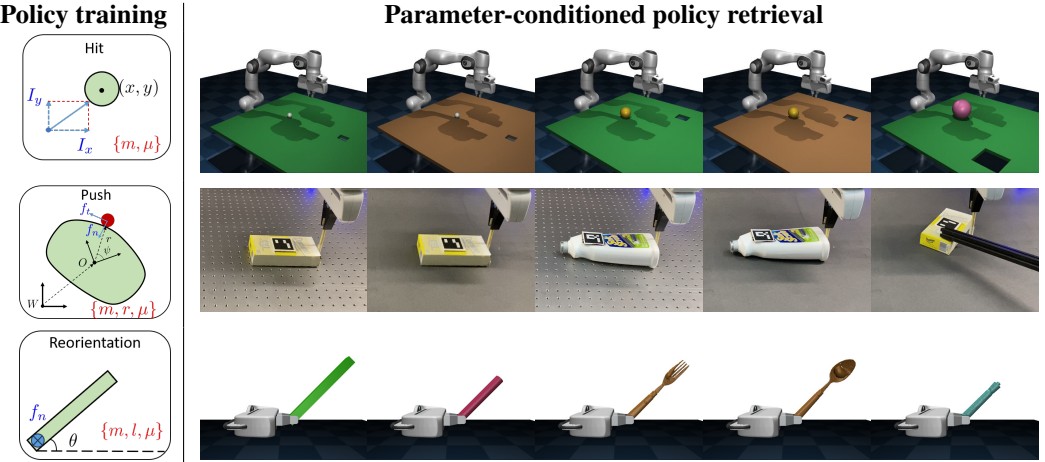

Figure 1: Overview of the proposed bi-level approach. **Left: Parameter-augmented policy training using multiple models.** The state, action, and parameter variables are denoted in black, blue, and red colors, respectively. **Right: Parameter-conditioned policy retrieval through domain contraction.** The retrieved policies perform well in terms of both generalization and optimality given a diverse set of objects with different shapes, weights, and friction parameters.

**Abstract:** Contact-rich manipulation plays an important role in human daily activities, but uncertain parameters pose significant challenges for robots to achieve comparable performance through planning and control. To address this issue, domain adaptation and domain randomization have been proposed for robust policy learning. However, they either lose the generalization ability across diverse instances or perform conservatively due to neglecting instance-specific information. In this paper, we propose a bi-level approach to learn robust manipulation primitives, including parameter-augmented policy learning using multiple models, and parameter-conditioned policy retrieval through domain contraction. This approach unifies domain randomization and domain adaptation, providing optimal behaviors while keeping generalization ability. We validate the proposed method on three contact-rich manipulation primitives: hitting, pushing, and reorientation. The experimental results showcase the superior performance of our approach in generating robust policies for instances with diverse physical parameters.

**Keywords:** Robust policy learning, Contact-rich manipulation, Sim-to-real

8th Conference on Robot Learning (CoRL 2024), Munich, Germany.

# 1  Introduction

Robot manipulation usually involves multiple different manipulation primitives, such as `Push` and `Pivot`, leading to hybrid and long-horizon characteristics. This poses significant challenges to most planning and control approaches. Instead of treating long-horizon manipulation as a whole, it can be decomposed into several simple manipulation primitives and then sequenced using PDDL planners [1, 2, 3] or Large Language Models [4, 5]. Although such manipulation primitives usually have low-to-medium-dimensional state and action spaces, the breaking and establishment of contact make it tough for most motion planning techniques. Gradient-based techniques suffer from vanishing gradients when contact breaks, while sampling-based techniques struggle with the combinatorial complexity of multiple contact modes, i.e., sticking and sliding. This leads to time-consuming online replanning in the real world for contact-rich manipulation, limiting the real-time reactiveness of robots in coping with uncertainties and disturbances.

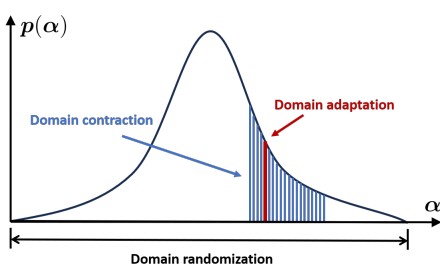

Figure 2: Illustration of DA, DR and DC. $\alpha$ is the physical parameter. $p(\alpha)$ is the corresponding probability. DR conditions on the entire domain, DA on a specific parameter (red line), and DC on a small set (blue lines).

Learning manipulation primitives that can quickly react to the surroundings, therefore, makes a lot of sense. Since the learned manipulation primitives will be sequenced by symbolic planners, which have no information about the geometric/motion level, the learned manipulation primitive should be robust to diverse instances with varied physical parameters, such as shape, mass, and friction coefficient. For example, once the `push` primitive is scheduled by the high-level symbolic planner, it should be able to push any objects with different shapes and friction parameters from any initial configurations towards their targets.

Consider how humans perform manipulation tasks: we typically make rough estimates of object and environmental parameters based on physical intuition [6], and our behaviors adapt accordingly. To enable robots to achieve similar performance, the learned manipulation primitives should effectively generalize across diverse instances and adapt to the parameters of specific instances. Common techniques for robust policy learning include domain adaptation (DA) [7, 8, 9] and domain randomization (DR) [10, 11]. DA aims for a perfect match between simulation and reality but may sacrifice generalization ability, while DR generalizes well but lacks full knowledge of the target domain, leading to conservative behaviors. Combining DR and DA [11, 12] to harness the benefits of both is a sensible strategy, but how to effectively balance them is still an open question. In this work, we propose a bi-level approach to address this challenge:

**Level-1: Parameter-augmented policy learning using multiple models.** We augment the state space with physical parameters of multiple models and use Tensor Train (TT) to approximate the state-value function and advantage function.

**Level-2: Parameter-conditioned policy retrieval through domain contraction.** At the stage of execution, we can obtain a rough estimate of the physical parameters in the manipulation domain. This instance-specific information can be utilized to retrieve a parameter-conditioned policy, expected to be much more instance-optimal.

We summarize our contributions as follows:

1) We propose domain contraction, a unification of domain adaptation and domain randomization, which results in optimal behavior while maintaining generalization.

2) We propose using tensor approximation for robust policy learning with multiple models and leveraging products of tensor cores for parameter-conditioned policy retrieval.

3) We provide both theoretical proof and numerical comparisons for the proposed approach.

## 2 Related Work

**Manipulation policy learning.** Many approaches have been proposed to learn manipulation policies for long-horizon manipulation, including Behavior Cloning (BC) [13, 14], Deep Reinforcement Learning (DRL) [15, 12, 16], and Approximate Dynamic Programming (ADP) [17, 18]. In our work, to ensure subsequent parameter-conditioned policy retrieval, it is essential to explore the full parameter domain. The reliance on the provided dataset in BC limits its ability to explore the broad state space and learn the full parameter-augmented policy. DRL offers better exploration capability compared to BC, but its policy training is usually time-consuming due to sample inefficiency, and policy retrieval is often suboptimal due to gradient-based optimization and local exploration. Conversely, ADP aims to cover the full state space but faces the curse of dimensionality. Recently, a new technique called Tensor Train Policy Iteration (TTPI) [19] has emerged to address this challenge by employing tensor approximation. Our work builds upon TTPI, extending it for robust primitive learning by augmenting the state space with model parameters.

**Sim-to-real transfer.** Obtaining large amounts of real-world data for primitive learning is challenging. Therefore, robot learning in simulation and transferring to the real world is a promising idea [20]. However, the reality gap between simulation and the real world poses a significant challenge to such idea. If the target domain is known and specific, Domain Adaptation (DA) [7, 8, 9] can be an effective method, but its reliance on domain-specific data can limit generalization to other scenarios without additional fine-tuning. On the other hand, Domain Randomization (DR) [21] seeks to develop a robust policy by introducing random variations into the simulation parameters. While this method offers good generalization, it often results in suboptimal and high-variance behaviors due to the restrictive assumptions about environment parameter distribution (e.g., normal or uniform). To overcome this, combining DA and DR can be a promising strategy to balance generalization and optimal performance. The basic idea is to adapt the parameter distribution by leveraging differences between simulated and reference environment [10, 11, 22]. However, policies developed through such methods are often tailored to the reference environment or target domain. Recently, a method called Rapid Motor Adaptation (RMA) [23] has demonstrated remarkable success in learning robust locomotion [16, 24] and manipulation primitives [12]. In this approach, a teacher policy is trained in simulation with various domain parameters, and a student policy learns to replicate this behavior, with a low-dimensional embedding of proprioceptive history as input. Our work closely follows this paradigm but introduces a more efficient way for learning the parameter-augmented teacher policy. Additionally, our method eliminates the need to learn a student policy afterward, as the parameter-conditioned robust policy can be easily retrieved from the teacher policy. It also offers a flexible way to leverage domain knowledge, making it effective in both closed-loop and open-loop scenarios, whereas RMA struggle in open-loop scenarios due to the lack of privileged information.

**Tensor Train for function approximation.** A multidimensional function can be approximated by a tensor, where each element in the tensor is the value of the function given the discretized inputs. The continuous value of the function can then be obtained by interpolating among tensor elements. However, storing the full tensor for a high-dimensional function can be challenging. To address this issue, Tensor Train (TT) was proposed to approximate the tensor using several third-order cores. The widely used methods include TT-SVD [25] and TT-cross [26]. Furthermore, TTGO [27] was proposed for finding globally optimal solutions given functions in TT format. Thanks to the low-rank embeddings of the original function, this method significantly enhances computation efficiency and reduces the risk of getting trapped in local optima. TTPI [19] was then introduced to learn control policies through tensor approximation, showing superior performance on several hybrid control problems. Logic-Skill Programming (LSP) [2] expands the operational space of TTPI by incorporating first-order logic to sequence policies. Building on TTGO and TTPI, our work extends these methods to learn robust policies, which further enhances the capabilities of LSP. We additionally demonstrate that the TT format is a suitable structure for domain contraction, allowing for efficient parameter-conditioned policy retrieval.

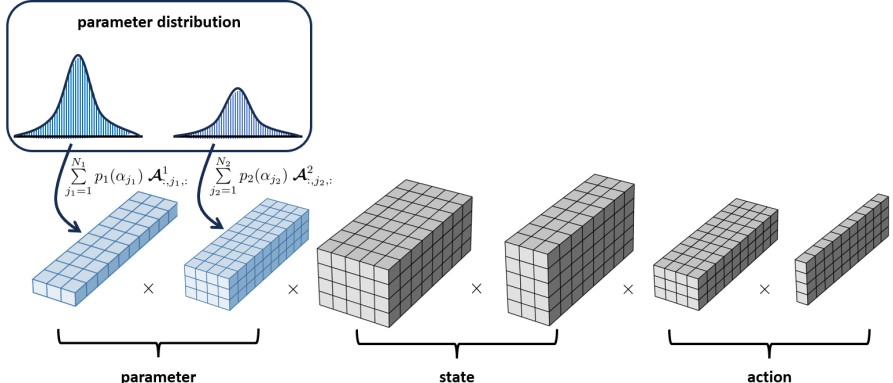

Figure 3: Overview of domain contraction in TT format. Using TTPI, we can obtain the parameter-augmented advantage function in TT format. It includes separate 3rd-order cores for different dimensionality, such as parameter, state and action. In this figure, we demonstrate the advantage function for `Hit` primitive. Given the parameter distribution (either by human knowledge or by system identification), we can retrieve the parameter-conditioned policy by making product of parameter distributions and corresponding TT cores.

## 3 Method

### 3.1 Problem formulation

Consider a discrete-time dynamical system defined by:

$$\boldsymbol{x}_{t+1} \sim f(\boldsymbol{x}_t, \boldsymbol{u}_t | \boldsymbol{\alpha}), \quad \boldsymbol{u}_t \sim \pi_\theta(\boldsymbol{x}_t | \boldsymbol{\alpha}), \quad \boldsymbol{\alpha} \sim \boldsymbol{p}(\boldsymbol{\alpha}), \tag{1}$$

where $\boldsymbol{x}_t \in \Omega_{\boldsymbol{x}}$ and $\boldsymbol{u}_t \in \Omega_{\boldsymbol{u}}$ represent the state and action at time step $t$, respectively. The dynamics model $f : \Omega_{\boldsymbol{x}} \times \Omega_{\boldsymbol{u}} \times \Omega_{\boldsymbol{\alpha}} \to \Omega_{\boldsymbol{x}}$ is conditioned on the domain (or environment) parameters $\boldsymbol{\alpha} \in \Omega_{\boldsymbol{\alpha}}$ (such as masses, shapes, or friction coefficients). These parameters are assumed to be random variables satisfying an unknown probability distribution $\boldsymbol{p}$. Together with the reward function $R : \Omega_{\boldsymbol{x}} \times \Omega_{\boldsymbol{u}} \to \mathbb{R}$ and the discount factor $\gamma \in [0, 1]$, this system forms a Markov Decision Process (MDP) $\mathcal{M} = \{\Omega_{\boldsymbol{x}}, \Omega_{\boldsymbol{u}}, \Omega_{\boldsymbol{\alpha}}, \boldsymbol{p}, f, R, \gamma\}$.

In this work, we aim to find the optimal policy that maximizes the expected cumulative reward for a distribution of domain parameters $\boldsymbol{\alpha} \sim \boldsymbol{p}(\boldsymbol{\alpha})$, namely

$$V(\boldsymbol{x}|\boldsymbol{\alpha}) = \mathbb{E}_{\boldsymbol{\alpha} \sim \boldsymbol{p}(\boldsymbol{\alpha})} \left[ \mathbb{E}_\pi \left[ \sum_{t=0}^{\infty} \gamma^t R(\boldsymbol{x}_t, \boldsymbol{u}_t) \mid \boldsymbol{x}_0 = \boldsymbol{x} \right] \right],$$

$$A(\boldsymbol{x}, \boldsymbol{u}|\boldsymbol{\alpha}) = R(\boldsymbol{x}, \boldsymbol{u}) + \gamma(V(f(\boldsymbol{x}, \boldsymbol{u}|\boldsymbol{\alpha})) - V(\boldsymbol{x}|\boldsymbol{\alpha})), \quad \pi(\boldsymbol{x}|\boldsymbol{\alpha}) = \arg \max_{\boldsymbol{u} \in \Omega_{\boldsymbol{u}}} A(\boldsymbol{x}, \boldsymbol{u}|\boldsymbol{\alpha}). \tag{2}$$

To learn such parameter-conditioned policies, it is impractical to train them individually since there can be infinite instances for one single primitive. Instead, we propose to learn a parameter-augmented full policy first and then retrieve the parameter-conditioned optimal policy for specific domain at runtime.

### 3.2 Robust policy learning through domain contraction

We define domain contraction as retrieving the parameter-conditioned policy from the weighted sum of parameter-specific advantage functions. The theoretical proof is provided in Sec. 3.3. These functions are learned jointly by augmenting the state space with parameters, similar to the multi-goal setting in DRL [28], resulting in parameter-augmented advantage function.

TTPI [19] is used to approximate the parameter-augmented value function

$$V(\boldsymbol{\alpha}, \boldsymbol{x}) \approx \boldsymbol{\mathcal{V}}(\alpha_{1:d}, \boldsymbol{x}_{1:m}) = \boldsymbol{\mathcal{V}}^1_{:,i_1,:} \cdots \boldsymbol{\mathcal{V}}^d_{:,i_d,:} \boldsymbol{\mathcal{V}}^{d+1}_{:,i_{d+1},:} \cdots \boldsymbol{\mathcal{V}}^{d+m}_{:,i_{d+m},:} \tag{3}$$

and the parameter-augmented advantage function

$$A(\boldsymbol{\alpha}, \boldsymbol{x}, \boldsymbol{u}) \approx \mathcal{A}(\alpha_{1:d}, \boldsymbol{x}_{1:m}, \boldsymbol{u}_{1:n})$$
$$= \mathcal{A}^1_{:,i_1,:} \cdots \mathcal{A}^d_{:,i_d,:} \, \mathcal{A}^{d+1}_{:,i_{d+1},:} \cdots \mathcal{A}^{d+m}_{:,i_{d+m},:} \, \mathcal{A}^{d+m+1}_{:,i_{d+m+1},:} \cdots \mathcal{A}^{d+m+n}_{:,i_{d+m+n},:} \quad (4)$$

in TT format. We define the tensor cores related to $\boldsymbol{x}$ and $\boldsymbol{u}$ as

$$\mathcal{A}(\boldsymbol{x}_{1:m}, \boldsymbol{u}_{1:n}) = \mathcal{A}^{d+1}_{:,i_{d+1},:} \cdots \mathcal{A}^{d+m}_{:,i_{d+m},:} \, \mathcal{A}^{d+m+1}_{:,i_{d+m+1},:} \cdots \mathcal{A}^{d+m+n}_{:,i_{d+m+n},:} \quad (5)$$

Without loss of generality, we assume each subspace $\Omega_{\alpha_i}$ of the parameter space $\Omega_\alpha = \Omega_{\alpha_1} \times \cdots \times \Omega_{\alpha_d}$ is discretized by $N_i$ points. The parameter-specific advantage function can then be extracted as

$$A_{\alpha_j}(\boldsymbol{x}, \boldsymbol{u}) \approx \mathcal{A}(\boldsymbol{x}_{1:m}, \boldsymbol{u}_{1:n}|\boldsymbol{\alpha}_j) = \mathcal{A}^1_{:,j_1,:} \cdots \mathcal{A}^d_{:,j_d,:} \, \mathcal{A}(\boldsymbol{x}_{1:m}, \boldsymbol{u}_{1:n}), \quad (6)$$

where $\boldsymbol{\alpha}_j = (\alpha_{j_1}, \cdots, \alpha_{j_d})$ represents $\boldsymbol{\alpha}$ at the discretization index $j$ across all dimensions.

We define $p_j = p_1(\alpha_{j_1}) p_2(\alpha_{j_2}) \cdots p_d(\alpha_{j_d})$ as the probability of $\boldsymbol{\alpha}_j$, and $p_i$ is the probability distribution at dimension $i$. Given the TT approximation $\mathcal{P}_{(j_1,\ldots,j_d)} = \mathcal{P}^1_{:,j_1,:} \mathcal{P}^2_{:,j_2,:} \cdots \mathcal{P}^d_{:,j_d,:}$ of the rough parameter distribution $p_j$, we can then compute the parameter-conditioned advantage function by computing the weighted sum of parameter-specific advantage functions in a separable form

$$A(\boldsymbol{x}, \boldsymbol{u}|\boldsymbol{\alpha}) = \sum_{j_1=1}^{N_1} \cdots \sum_{j_d=1}^{N_d} p_j A_{\alpha_j}(\boldsymbol{x}, \boldsymbol{u})$$

$$\approx \sum_{j_1=1}^{N_1} \mathcal{P}^1_{:,j_1,:} \, \mathcal{A}^1_{:,j_1,:} \cdots \sum_{j_d=1}^{N_d} \mathcal{P}^d_{:,j_d,:} \, \mathcal{A}^d_{:,j_d,:} \, \mathcal{A}(\boldsymbol{x}_{1:m}, \boldsymbol{u}_{1:n}). \quad (7)$$

The computation of weighted sum is at the tensor core level, which is much more efficient than at the function level. After obtaining the parameter-conditioned advantage function, we can retrieve the parameter-conditioned policy using TTGO [27], a specialized method for efficiently finding globally optimal solutions given functions in TT format.

To retrieve such parameter-conditioned policies, knowing the parameter distribution $\boldsymbol{p}$ is crucial. DR and DA rely on assumed distributions during training, while DC offers a better way to leverage domain knowledge during execution, enabling optimal behaviors while preserving generalization capability. In Section 3.3, we demonstrate that DR and DA are two special cases of DC.

## 3.3 Theoretical proof

**Theorem 1.** *Given domain parameters $\boldsymbol{\alpha}$ and the distribution $\boldsymbol{p}$, the parameter-conditioned policy can be retrieved from the weighted sum of parameter-specific advantage functions.*

*Proof.* Given that each subspace $\Omega_{\alpha_i}$ of the parameter space $\Omega_\alpha$ is discretized by $N_i$ points, the parameter-conditioned policy can be written as

$$A(\boldsymbol{x}, \boldsymbol{u}|\boldsymbol{\alpha}) = R(\boldsymbol{x}, \boldsymbol{u}) + \sum_{j_1=1}^{N_1} \cdots \sum_{j_d=1}^{N_d} p_j \, \gamma\big(V\big(f(\boldsymbol{x}, \boldsymbol{u}|\boldsymbol{\alpha}_j)\big) - V(\boldsymbol{x}|\boldsymbol{\alpha}_j)\big),$$

$$= \sum_{j_1=1}^{N_1} \cdots \sum_{j_d=1}^{N_d} p_j \Big( R(\boldsymbol{x}, \boldsymbol{u}) + \gamma\big(V\big(f(\boldsymbol{x}, \boldsymbol{u}|\boldsymbol{\alpha}_j)\big) - V(\boldsymbol{x}|\boldsymbol{\alpha}_j)\big)\Big), \quad (8)$$

and the parameter-specific advantage function is defined as

$$A_{\alpha_j}(\boldsymbol{x}, \boldsymbol{u}) = R(\boldsymbol{x}, \boldsymbol{u}) + \gamma\big(V\big(f(\boldsymbol{x}, \boldsymbol{u}|\boldsymbol{\alpha}_j)\big) - V(\boldsymbol{x}|\boldsymbol{\alpha}_j)\big). \quad (9)$$

Given (8) and (9), we can derive that the parameter-conditioned advantage function is the weighted sum of parameter-specific advantage functions, namely

$$A(\boldsymbol{x}, \boldsymbol{u}|\boldsymbol{\alpha}) = \sum_{j_1=1}^{N_1} \cdots \sum_{j_d=1}^{N_d} p_j A_{\alpha_j}(\boldsymbol{x}, \boldsymbol{u}), \quad (10)$$

and the parameter-conditioned primitive policy can then be computed by

$$\pi(\boldsymbol{x}|\boldsymbol{\alpha}) = \arg \max_{\boldsymbol{u} \in \Omega_{\boldsymbol{u}}} A(\boldsymbol{x}, \boldsymbol{u}|\boldsymbol{\alpha}). \tag{11}$$

Note that the parameter-conditioned policy cannot be directly computed as the weighted sum of parameter-specific policies, since the $\arg \max$ operation does not have an associative property with respect to addition.

**Theorem 2.** *Domain randomization and domain adaptation are two special cases of domain contraction.*

*Proof.* The policies of DR, DA and DC are obtained by

$$\pi_{\mathrm{DR}}(\boldsymbol{x}) = \arg \max_{\boldsymbol{u} \in \Omega_{\boldsymbol{u}}} \sum_{j_1=1}^{N_1} \cdots \sum_{j_d=1}^{N_d} \widetilde{p}_j A_{\alpha_j}(\boldsymbol{x}, \boldsymbol{u}), \quad \pi_{\mathrm{DA}}(\boldsymbol{x}) = \arg \max_{\boldsymbol{u} \in \Omega_{\boldsymbol{u}}} \sum_{j_1=1}^{N_1} \cdots \sum_{j_d=1}^{N_d} \overline{p}_j A_{\alpha_j}(\boldsymbol{x}, \boldsymbol{u}),$$

$$\pi_{\mathrm{DC}}(\boldsymbol{x}) = \arg \max_{\boldsymbol{u} \in \Omega_{\boldsymbol{u}}} \sum_{j_1=1}^{N_1} \cdots \sum_{j_d=1}^{N_d} p_j A_{\alpha_j}(\boldsymbol{x}, \boldsymbol{u}).$$

where $\widetilde{p}_j$, $\overline{p}_j$ and $p_j$ are the probabilities of the parameter instance $\boldsymbol{\alpha}$ at index $j$, satisfying the distributions $\widetilde{\boldsymbol{p}}$, $\overline{\boldsymbol{p}}$ and $\boldsymbol{p}$, which are used for policy retrieval in DR, DA and DC, respectively.

Note that at the policy training level, we have no information about the real distribution $\boldsymbol{p}^*$. DR assumes $\boldsymbol{p}^*$ as a uniform or Gaussian distribution $\widetilde{\boldsymbol{p}}$ (which can be a wrong assumption) for policy training. DA aims for closed match between simulation and the target domain by using partly the real data, resulting in $\overline{\boldsymbol{p}} \approx \boldsymbol{p}^*$. Conversely, DC enables the use of domain knowledge during the execution stage, where instance-specific information becomes available. We assume the estimated rough parameter distribution as $\boldsymbol{p}$. If $\boldsymbol{p}$ matches $\boldsymbol{p}^*$, the policy aligns with DA; if no instance knowledge is present, $\boldsymbol{p}$ equals $\widetilde{\boldsymbol{p}}$, resulting in a DR-like policy. DC therefore unifies DA and DR, while allowing flexible compromise between these two extreme cases.

## 4   Experimental Results

We validate the effectiveness of the proposed method on three contact-rich manipulation tasks: `Hit`, `Push`, and `Reorientation`, as shown in Fig. 1. All the primitives are highly dependent on the physical parameters between the object, the robot, and the surroundings. The environmental details, including the task setup and implementation details, are presented in the appendix.

### 4.1   Robust policy learning and retrieval

We first learn the parameter-augmented policies using multiple models through tensor approximation. The physical equation of `Hit` is fully known, we can therefore compute the control policy analytically. For `Push` and `Reorientation`, TTPI is used to approximate the parameter-augmented value functions and advantage functions.

**Hit**: Given the mass $m$, the friction coefficient $\mu$, the initial state $\boldsymbol{x}_0$ and the target $\boldsymbol{x}^{\mathrm{des}}$, the advantage function is

$$A(\boldsymbol{x}, \boldsymbol{I}) = -\big(\|\boldsymbol{x} - \boldsymbol{x}^{\mathrm{des}}\|^2 + 0.01\|\boldsymbol{I}\|^2\big), \tag{12}$$

where $\boldsymbol{x} = \boldsymbol{x}_0 + \frac{\boldsymbol{I}}{m}t - 0.5\mu g t^2$. Computing the impact $\boldsymbol{I}$ is to find the value that maximizes (12).

**Push**: The reward function for `Push` primitive learning is defined as

$$r = -(\rho c_p + c_o + 0.01 c_f + 0.01 c_v), \tag{13}$$

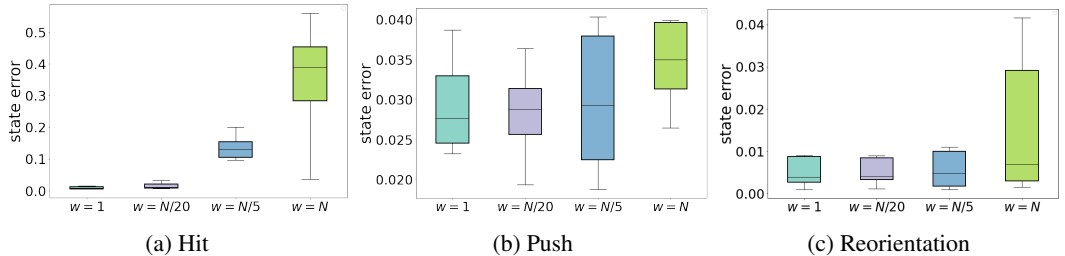

|  | (a) Hit | (b) Push | (c) Reorientation |

Figure 4: Comparison of final state error given different estimated parameter distributions

with

$$c_p = \|\boldsymbol{x}_p - \boldsymbol{x}_p^{\text{des}}\|/l_p, \quad c_o = \|x_o - x_o^{\text{des}}\|/l_o, \quad c_f = \|\boldsymbol{f}\|, \quad c_v = \|\boldsymbol{v}_p\|, \tag{14}$$

where $\boldsymbol{x}_p = [s_x, s_y]$ and $x_o = \theta$ denote the object's position and orientation. Without loss of generality, we set $\boldsymbol{x}^{\text{des}} = \boldsymbol{0}$ as the target configuration. $\boldsymbol{f}$ is the control force, while $\boldsymbol{v}_p$ is the velocity of the robot's end-effector. $l_p$ and $l_o$ are set to $0.005$ and $0.01\pi$, respectively.

**Reorientation**: The reward function for `Reorientation` primitive learning is defined as

$$r = -(\beta c_g + c_f), \tag{15}$$

with $c_g = \|x_o - x_o^{\text{des}}\|$, $c_f = \|f_n\|$. $x_o$ is the orientation angle $\theta$, and $f_n$ is the normal force between the gripper and object. $x_o^{\text{des}}$ is set to $\pi$ as the reorientation goal, and $\beta$ is set to $10^4$.

We then retrieve the parameter-conditioned policy through domain contraction. Note that any distribution of the parameters is allowed in our framework. Without loss of generality, we assume the parameters satisfy a uniform distribution, within a range of the discretization indices of each dimensionality (denoted as $w$), as shown in Table 1 and Fig. 4. $w = 1$ means we know the exact value of the physical parameter, corresponding to domain adaptation. In contrast, $w = N$ means we have no prior knowledge about the model parameters, corresponding to domain randomization.

Table 1: Cumulative reward of three manipulation tasks

|  | $w = 1$ | $w = N/20$ | $w = N/5$ | $w = N$ |
|---|---|---|---|---|
| Hit | 1.0 | $0.65 \pm 0.21$ | $0.01 \pm 0.01$ | $0.02 \pm 0.05$ |
| Push | 1.0 | $0.99 \pm 0.01$ | $0.99 \pm 0.03$ | $0.93 \pm 0.11$ |
| Reori. | 1.0 | $0.99 \pm 0.04$ | $0.99 \pm 0.07$ | $0.85 \pm 0.19$ |

Table 1 and Fig. 4 demonstrate the comparisons of cumulative reward and final state error, respectively. The state error is quantified as the L2 norm of the difference between the final state and the target state. The cumulative reward is normalized using the value obtained through domain adaptation. We can observe that `Hit` primitive depends more on the accuracy of model parameters, as it resembles an open-loop control. Once the impact is given from the robot to the object, there is no way to adjust control inputs to influence the object movements further. However, as the comparison indicates, there is no need to have a precise parameter estimation. A rough range ($w = N/20$) is sufficient to achieve the target. In contrast, `Push` and `Reorientation` can be considered more akin to closed-loop control. Therefore, the requirement for accurate parameter estimation can be relaxed further. As depicted in Table 1 and Fig. 4, a rough distribution with $w = N/5$ is adequate. Moreover, based on Table 1, we observe that $w = N$ results in the lowest cumulative reward. This is consistent with our assertion that domain randomization typically leads to conservative behaviors. Although domain adaptation ($w = 1$) yields the highest cumulative reward, obtaining precise parameter values can be challenging in the real world. Domain contraction bridges the gap between domain adaptation and domain randomization, offering greater flexibility to generate optimal behaviors while utilizing instance-specific rough parameter distribution. This is much practical for real-world contact-rich manipulation tasks.

## 4.2 Comparison to other primitive learning methods

Table 2: Comparison of robust primitive learning approaches (time in [s]econd or [m]inute)

| | RL+DR | | RMA | | | Ours | | |
|---|---|---|---|---|---|---|---|---|
| | time | error | time | teacher_error | student_error | time | teacher_error | DC_error |
| Hit | 3.76s | 0.653 ± 0.53 | 4.56s | 0.036 ± 0.030 | NA | **0.20s** | **0.011 ± 0.007** | **0.015 ± 0.010** |
| Push | 41.92m | 0.061 ± 0.003 | 54.72m | 0.053 ± 0.103 | 0.039 ± 0.126 | **5.83m** | **0.035 ± 0.023** | **0.036 ± 0.024** |
| Reori. | 21.78m | 0.034 ± 0.026 | 23.23m | 0.086 ± 0.065 | 0.104 ± 0.087 | **1.32m** | **0.010 ± 0.014** | **0.011 ± 0.014** |

Moreover, we compared our method with two widely used robust policy learning approaches: reinforcement learning with domain randomization (RL+DR) [21] and rapid motor adaptation (RMA) [23, 12]. The quantitative results are presented in Table 2, including the time required for policy training and the final state error. Note that in RMA, **teacher_error** and **student_error** correspond to the errors of the teacher policy given the true parameter and the student policy, respectively, while in our method, **teacher_error** and **DC_error** correspond to the errors of the parameter-augmented policy given the true parameter and the parameter-conditioned policy obtained through domain contraction (DC). Our method requires significantly less time for policy training and results in much lower final state error compared to the other methods. Furthermore, when comparing the error of the teacher policy in RMA with our parameter-augmented policy, our method shows better accuracy, highlighting its potential as a teacher policy in the general RMA framework. Additionally, in one-shot manipulation tasks where only one action can be executed (as shown in the `Hit` primitive in the table), RMA does not work due to the lack of privileged information.

### 4.3 Real-robot experiments: planar push

We validated our proposed method for the planar pushing task using a 7-axis Franka robot and a RealSense D435 camera. The manipulated objects included a sugar box and a bleach cleanser from the YCB dataset [29], each with different shapes and masses, as shown in Fig. 1. The friction coefficients between the objects and the table were varied by using a metal surface and plywood, respectively. Note that it is easier to control the robot kinematically rather than using force control. We leverage the ellipsoidal limit surface to convert the applied force to velocity, resulting in the motion equations shown in [30, 31]. We trained a parameter-augmented policy in simulation and then applied it in the real world through domain contraction. The experimental results showcase the effectiveness of the obtained parameter-conditioned policies in manipulating instances with diverse parameters. Additionally, external disturbances were introduced by humans, demonstrating the reactiveness of the retrieved policy. Additional results are presented in the accompanying video.

## 5 Conclusion and Future Work

In this paper, we propose a bi-level approach to learning robust manipulation primitives. Multiple models and tensor approximation are used to train the parameter-augmented policy, which can be directly applied to diverse instances through domain contraction, outputting instance-dependent optimal behaviors. Theoretical proof and numerical results demonstrate the efficiency of the proposed method for robust primitive learning. Real-world experiments further validate its effectiveness in contact-rich manipulation under uncertainty and disturbance.

Since we leverage TTPI to learn the parameter-augmented policy, we also adopt its limitations. TTPI assumes a low-rank structure of the value function and advantage function. Therefore, the state space should have low-to-medium dimensionality (approximately less than 35), limiting its application for image-based policy learning. In the future, this limitation can be addressed by combining TT with neural networks in a data-driven manner [32, 33].

Moreover, we assume that rough distributions of uncertain physical parameters are provided by human knowledge. This requirement can be eliminated by integrating Large Visual Language Models or system identification into the framework. As shown in our experiments, the parameter estimation does not need to be very precise, thanks to the proposed domain contraction technique. Additionally, the policy can be iteratively improved as the belief about parameter estimation is updated.

**Acknowledgments**

This work was supported by the China Scholarship Council (grant No.202106230104), and by the SWITCH project (https://switch-project.github.io/), funded by the Swiss National Science Foundation. We thank Jiacheng Qiu for suggestions about the implementation of RL baselines.

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

# A    Background of Tensor Train

## A.1    Tensors as Discrete Analogue of a Function

A multivariate function $P(x_1, \ldots, x_d)$ defined over a rectangular domain constructed with the Cartesian product of intervals (or discrete sets) $I_1 \times \cdots \times I_d$ can be discretized by evaluating it at points in the set $\mathcal{X} = \{(x_1^{i_1}, \ldots, x_d^{i_d}) : x_k^{i_k} \in I_k, i_k \in \{1, \ldots, n_k\}\}$. This gives us a tensor $\mathcal{P}$, a discrete version of $P$, where $\mathcal{P}_{(i_1, \ldots, i_d)} = P(x_1^{i_1}, \ldots, x_d^{i_d}), \forall (i_1, \ldots, i_d) \in \mathcal{I}_{\mathcal{X}}$, and $\mathcal{I}_{\mathcal{X}} = \{(i_1, \ldots, i_d) : i_k \in \{1, \ldots, n_k\}, k \in \{1, \ldots, d\}\}$. The value of $P$ at any point in the domain can then be approximated by interpolating between the elements of the tensor $\mathcal{P}$.

## A.2    Tensor Networks and Tensor Train Decomposition

Naively approximating a high-dimensional function using a tensor is intractable due to the combinatorial and storage complexities of the tensor ($\mathcal{O}(n^d)$). Tensor networks mitigate the storage issue by decomposing the tensor into factors with fewer elements, akin to using Singular Value Decomposition (SVD) to represent a large matrix. In this paper, we explore the use of Tensor Train (TT), a type of Tensor Network that represents a high-dimensional tensor using several third-order tensors called *cores*, as shown in Fig. 5.

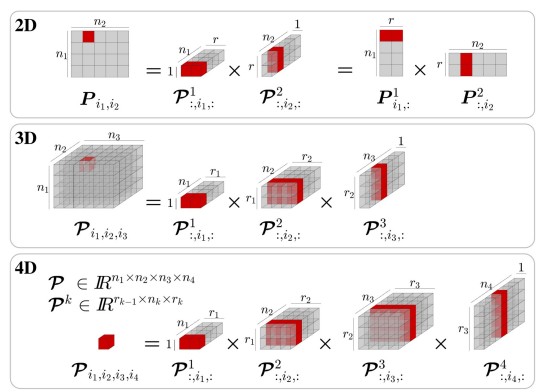

Figure 5: TT decomposition generalizes matrix decomposition techniques to higher-dimensional arrays. In TT format, an element in a tensor can be obtained by multiplying specific slices of the core tensors. The figure presents examples of second-order, third-order, and fourth-order tensors. Image adapted from [27].

We can access the element $(i_1, \ldots, i_d)$ of the tensor in this format simply given by multiplying matrix slices from the cores:

$$\mathcal{P}_{(i_1, \ldots, i_d)} = \mathcal{P}^1_{:,i_1,:} \mathcal{P}^2_{:,i_2,:} \cdots \mathcal{P}^d_{:,i_d,:}, \tag{16}$$

where $\mathcal{P}^k_{:,i_k,:} \in \mathbb{R}^{r_{k-1} \times r_k}$ represents the $i_k$-th frontal slice (a matrix) of the third-order tensor $\mathcal{P}^k$. For any given tensor, there always exists a TT decomposition [25]. This low-rank structure further facilitates sampling and optimization for robot planning and control.

There are several ways to acquire a TT model, including TT-SVD [25] and TT-Cross [34, 35]. TT-SVD extends the SVD decomposition from matrix level to a high-dimensional tensor level. However, it needs to store the full tensor first, which is impractical for high-dimensional functions. TT-Cross solves this issue by selectively evaluating the function $P$ on a subset of elements, avoiding the need to store the entire tensor.

## A.3    Function approximation using Tensor Train

Given the discrete analogue tensor $\mathcal{P}$ of a function $P$, we obtain the continuous approximation by spline-based interpolation of the TT cores corresponding to the continuous variables only. For example, we can use linear interpolation for the cores (i.e., between the matrix slices of the core) and define a matrix-valued function corresponding to each core $k \in \{1, \ldots, d\}$,

$$\boldsymbol{P}^k(x_k) = \frac{x_k - x_k^{i_k}}{x_k^{i_k+1} - x_k^{i_k}} \mathcal{P}^k_{:,i_k+1,:} + \frac{x_k^{i_k+1} - x_k}{x_k^{i_k+1} - x_k^{i_k}} \mathcal{P}^k_{:,i_k,:}, \tag{17}$$

where $x_k^{i_k} \leq x_k \leq x_k^{i_k+1}$ and $\boldsymbol{P}^k : I_k \subset \mathbb{R} \to \mathbb{R}^{r_{k-1} \times r_k}$ with $r_0 = r_d = 1$. This induces a continuous approximation of $P$ given by

$$P(x_1, \ldots, x_d) \approx \boldsymbol{P}^1(x_1) \cdots \boldsymbol{P}^d(x_d). \tag{18}$$

This allows us to selectively do the interpolation only for the cores corresponding to continuous variables, and hence we can represent functions in TT format whose variables could be a mix of continuous and discrete elements.

### A.4 Global Optimization using Tensor Train (TTGO)

In optimization problems involving task parameters $\boldsymbol{x}$ and decision variables $\boldsymbol{u}$, the goal is to find the optimal $\boldsymbol{u}$ that minimizes the cost function $c(\boldsymbol{x}, \boldsymbol{u})$. TTGO [27] frames this problem as maximizing an unnormalized probability density function (PDF) $P(\boldsymbol{x}, \boldsymbol{u})$, which is derived from $c(\boldsymbol{x}, \boldsymbol{u})$ through a monotonically non-increasing transformation. For example, $P(\boldsymbol{x}, \boldsymbol{u})$ can be defined as $e^{-\beta C(\boldsymbol{x}, \boldsymbol{u})^2}$ with $\beta > 0$.

The TT-Cross algorithm is then used to compute the discrete analogue approximation of the unnormalized PDF, i.e., $\boldsymbol{\mathcal{P}}$, in the TT format. After approximating the joint distribution, $\boldsymbol{\mathcal{P}}^{\boldsymbol{x}_t}$ can be obtained by conditioning on the given task parameter $\boldsymbol{x} = \boldsymbol{x}_t \in \Omega_{\boldsymbol{x}}$.

Given the TT model $\boldsymbol{\mathcal{P}}^{\boldsymbol{x}_t}$, the optimal $\boldsymbol{u}$ is obtained through iterative sampling, which determines the optimal solution for each dimension at the core level. Due to the low-rank nature of the TT model, this sampling process is highly efficient. A number of prioritized samples, $N \geq 1$, are selected, and the sample(s) with the highest density (or lowest cost) are chosen as candidate solutions. These near-optimal solutions can then be further refined using local optimization methods, such as Newton-type optimization for continuous variables. Overall, this technique can efficiently find the globally optimal solution given any low-rank function, without requiring a convex structure. Moreover, the computation process is gradient-free and can handle a mix of continuous and discrete variables.

### A.5 Generalized Policy Iteration using Tensor Train (TTPI)

Optimal control of dynamic systems with nonlinear dynamics presents a significant challenge in robotics. To address this, Generalized Policy Iteration using Tensor Train (TTPI) was proposed, leveraging tensor approximation and approximate dynamic programming [19]. This method approximates state-value and advantage functions using Tensor Train (TT), effectively mitigating the curse of dimensionality. The low-rank structure of TT enables the use of TTGO to find near-global solutions during policy retrieval from the advantage function, even under complex nonlinear system constraints, surpassing the capabilities of existing neural network-based algorithms [33]. This approach does not require knowledge of the dynamics model; a black-box simulator suffices, akin to model-free reinforcement learning. Shetty et al. [19] demonstrated TTPI's superior performance on several hybrid control problems compared to state-of-the-art hybrid RL algorithms. In this paper, we apply this approach to learning manipulation primitives for contact-rich tasks involving many contact parameters. The typical TTPI algorithm cannot cope with the diverse contact-rich instances. Therefore, we propose domain contraction to retrieve the parameter-conditioned policy that can achieve robust manipulation.

## B Domain information for each manipulation primitive

### B.1 Hit

Hitting is widely used to manipulate objects through impact. In this work, we focus on planar hitting primitive. The state is the object position, denoted as $\boldsymbol{x} = [x, y]$. The control input is the applied impact $\boldsymbol{I} = [I_x, I_y]$. The physical parameters include the object mass $m$ and friction coefficient $\mu$. The motion equation is

$$\boldsymbol{x}^{\text{des}} = \boldsymbol{x}_0 + \frac{\boldsymbol{I}}{m}t - \frac{1}{2}\mu g t^2, \tag{19}$$

Given the initial state $\boldsymbol{x}_0$ and the target $\boldsymbol{x}^{\text{des}}$, the advantage function is

$$A(\boldsymbol{x}, \boldsymbol{I}) = -\left(\|\boldsymbol{x} - \boldsymbol{x}^{\text{des}}\|^2 + 0.01\|\boldsymbol{I}\|^2\right). \tag{20}$$

We applied TT to approximate the advantage function $A(\boldsymbol{x}, \boldsymbol{I})$, and the correct impact is computed by maximizing (20).

### B.2  Push

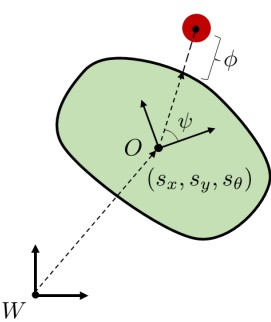

Figure 6: Illustration of pushing dynamics.

Pushing is challenging for robot planning and control due to its hybrid and under-actuated nature. The state is characterized by $[s_x, s_y, s_\theta, \psi, \phi]$, and the action is denoted as $[f_x, f_y, \dot{\psi}, \dot{\phi}]$. Here, $[s_x, s_y, s_\theta] \in SE(2)$ denotes the position and orientation of the object in the world frame. $\psi$ is the relative angle of the contact point in the object frame. $\phi$ represents the distance between the contact point and the object surface. $\boldsymbol{f} = [f_x, f_y]^\top$ are the forces exerted on the object, while $\boldsymbol{v}_p = [\dot{\psi}, \dot{\phi}]^\top$ represents the angular and translational velocities of the robot's end-effector. The physical parameters include the object mass $m$, radius $r$, and the friction coefficient $\mu$ between the object and table.

The applied force on this object can be mapped to its resulting velocity through a convex limit surface convex approximation [36], resulting in a sub-level set

$$H(\boldsymbol{w}) = \frac{1}{2}\boldsymbol{w}^\top \boldsymbol{L}\boldsymbol{w}, \tag{21}$$

where $\boldsymbol{L} = \text{diag}[f_{\max}^{-1}, f_{\max}^{-1}, m_{\max}^{-1}]$, with $f_{\max}$ as the maximum friction force between object and table, and $m_{\max}$ as the maximum torsional friction.

The robot dynamics is defined based on the Quasi-Static approximation and the limit surface, resulting in a similar expression as [37], namely

$$\dot{\boldsymbol{x}} = \begin{bmatrix} \boldsymbol{R}\boldsymbol{t} \\ \boldsymbol{v}_p \end{bmatrix} = \begin{bmatrix} \boldsymbol{R}\boldsymbol{L}\boldsymbol{w} \\ \boldsymbol{v}_p \end{bmatrix}, \tag{22}$$

with

$$\boldsymbol{R} = \begin{bmatrix} \cos\theta & -\sin\theta & 0 \\ \sin\theta & \cos\theta & 0 \\ 0 & 0 & 1 \end{bmatrix}, \tag{23}$$

$$\boldsymbol{w} = \begin{bmatrix} \boldsymbol{f} \\ \boldsymbol{\tau} \end{bmatrix} = \boldsymbol{J}^\top \boldsymbol{f} = \begin{bmatrix} 1 & 0 \\ 0 & 1 \\ -p_y & p_x \end{bmatrix} \boldsymbol{f}, \tag{24}$$

where $\boldsymbol{R}$ is the rotation matrix and $\boldsymbol{w}$ denotes the applied pusher wrench. $\boldsymbol{J}$ is the Jacobian matrix of the contact point in the body frame. The contact position $[p_x, p_y]^\top$ in the object frame can be computed by

$$p_x = \left(r(\psi) + \phi\right)\cos(\psi), \quad p_y = \left(r(\psi) + \phi\right)\sin(\psi), \tag{25}$$

for any shape that can be parameterized radially with the radial distance described as $r(\psi)$.

We parameterize the object shape using a concatenation of Bézier curves, with the weight matrix $C$ defined as

$$C = \begin{bmatrix} 1 & 0 & \cdots & 0 & 0 & \cdots & \cdots \\ 0 & 1 & \cdots & 0 & 0 & \cdots & \cdots \\ \vdots & \vdots & \ddots & \vdots & \vdots & \ddots & \cdots \\ 0 & 0 & \cdots & 1 & 0 & \cdots & \cdots \\ 0 & 0 & \cdots & 0 & 1 & \cdots & \cdots \\ 0 & 0 & \cdots & 0 & 1 & \cdots & \cdots \\ 0 & 0 & \cdots & 0 & 0 & \cdots & \cdots \\ \vdots & \vdots & \ddots & \vdots & \vdots & \ddots & \cdots \\ 0 & 0 & \cdots & \cdots & \cdots & 0 & 1 \\ 1 & 0 & \cdots & \cdots & \cdots & 0 & 0 \end{bmatrix}, \tag{26}$$

where the pattern $\begin{bmatrix} 1 & 0 \\ 0 & 1 \\ 0 & 1 \\ 0 & 0 \end{bmatrix}$ is repeated for each junction of two consecutive Bézier curves. For two concatenated cubic Bézier curves, each composed of 4 Bernstein basis functions, we can see locally that this operator yields a constraint of the form

$$\begin{bmatrix} w_3 \\ w_4 \\ w_5 \\ w_6 \end{bmatrix} = \begin{bmatrix} 1 & 0 \\ 0 & 1 \\ 0 & 1 \\ 0 & 0 \end{bmatrix} \begin{bmatrix} a \\ b \end{bmatrix}, \tag{27}$$

which ensures that $w_4 = w_5$. These constraints guarantee that the last control point and the first control point of the next segment are the same, therefore enforcing $C_0$ continuity of the reconstructed shape. Fig. 7 shows an example of reconstructing the shape of a mustard bottle from the YCB dataset.

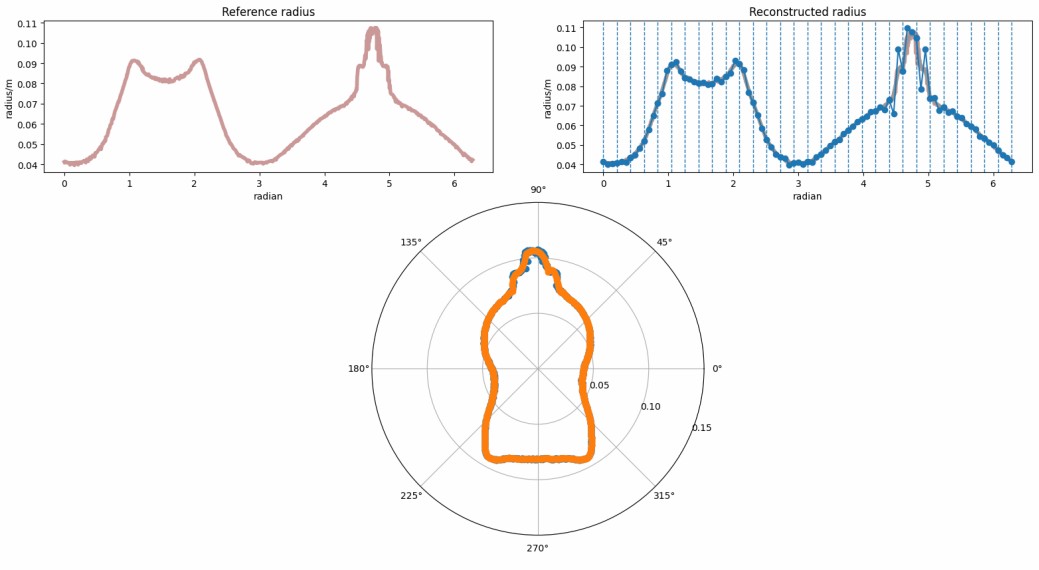

Figure 7: Shape parametrization of a mustard bottle using basis functions.

## B.3 Reorientation

In this task, we aim to enable the robot to reorient an object using parallel fingers. An initial velocity is given to the object by swinging the robot arm. The state is the orientation angle $\theta$, and the control input is the normal force $f_n$ between the gripper and object. The physical parameters are the object

Table 3: Time required for parameter-conditioned policy retrieval

|  | Policy retrieval (core-level) | Policy retrieval (function-level) |
|---|---|---|
| Hit | 0.016s $\pm$ 0.002s | 0.720s $\pm$ 0.236s |
| Push | 0.075s $\pm$ 0.003s | 18.56s $\pm$ 4.748s |
| Reorientation | 0.018s $\pm$ 0.003s | 8.494s $\pm$ 0.561s |

mass $m$, length $l$ and torsional friction coefficient $\mu$. The gravitational torque and normal force $f_n$ are used as braking mechanisms to slow down the object motion. We build the dynamics model of the reorientation primitive based on [38] as

$$
\begin{aligned}
I\ddot{\theta} &= \tau_g + 2\tau_f, \\
\dot{\theta} &= \dot{\theta}_0 - \ddot{\theta}\Delta t,
\end{aligned}
\tag{28}
$$

where $\tau_f = \mu_t f_n^{1+\gamma}$ is the torsional sliding friction between robot gripper and the object. In this work, we set $\gamma = 0$. $\mu_t$ is the torsional friction coefficient, which is related to the materials and normal force distribution. $\tau_g = mgl\sin(\theta)$ is the gravity torque. We therefore include $\mu_t$, object mass $m$ and length $l$ as the model parameters. The task is to rotate the object from a vertically downward to a vertically upward position. To achieve this, the object is given an initial angular velocity $\dot{\theta}_0$ by swinging the robot arm.

## C   Experimental details

### C.1   Implementation details

In our experiments, we employed an NVIDIA GeForce RTX 3090 GPU with 24GB of memory. For TTPI, the accuracy parameter was set to $\epsilon = 10^{-3}$ for TT-Cross approximation. The maximum rank $r_{\max}$ and discount factor were set to 100 and 0.99, respectively. The continuous variables of state, action and parameter domains were discretized as 50 to 500 points using uniform discretization.

The baseline algorithms, DC+RL and RMA, utilize Soft Actor-Critic (SAC) [39] for policy learning. Our implementation is based on Stable-Baselines3 [40], using a Multilayer Perceptron (MLP) with a $64 \times 64 \times 64$ architecture as the policy network. The discount factor is set to 0.99, and the learning rate to 0.001. Additionally, RMA employs an MLP with a $256 \times 128 \times 64$ architecture to embed privileged information.

### C.2   Extra experiments

We compared the time used for retrieving the parameter-conditioned policy in either core level or function level. Table 3 shows that TT structure allows for much more efficient retrieval via core-level products compared with function-level products.

