# OpenReview forum: "Robust Manipulation Primitive Learning via Domain Contraction"
_robot-learning.org/CoRL/2024/Conference — CoRL 2024_

### Official Review · Reviewer_aceS · 2024-07-18
**DC**

**Originality:** 2
**Technical Quality:** 3
**Clarity Of Presentation:** 1
**Potential Impact:** 2
**Recommendation:** 3
**Confidence:** 4

**Review:**

On the whole, the intuition and theoretical results seem solid, but the paper is very difficult to follow. Some additional text at the start of the method section should be added rather than immediately stating a theorem.  The problem should be stated clearly, i.e., we're in an MDP, the system dynamics are conditioned on the parameters $\alpha$, the goal is to recover the optimal policy $\pi^*$ for a specific distribution at runtime. Similarly, Theorem 2 regarding domain contraction is defined before providing a definition of DC. I would recommend stating a definition of the term and providing some intuition before proving its properties. And again, the definition of the various probability distributions is provided in the paragraph after the proof.  It feels like this whole section is backward from top to bottom. I would recommend introducing the problem, and at least define the parameter-conditioned policies and DC before providing their proofs. At present it is frankly difficult to follow without context or intuition.

Some related work in the context of adaptation for sim-to-real is notably omitted. For example, related work [1,2,3] discusses learning with privileged information to enable rapid adaptation at runtime, yet only vanilla DR and DA are discussed. It seems that these papers are actually the most closely related works. Other works, like BayesSim [4], similarly estimate a distribution of sim parameters given real data and retrieve policies robust to that distribution through RL training. I'm not entirely sure the notion of domain contraction is really novel.

The approach has some obvious limitations that are hinted at. The path toward high-dimensional observations is not obvious; the dynamics models need to be known a priori; the dynamic parameters need to be defined a priori. It isn't clear to me if this is a scalable approach or how it would be applied for more general, complex sim-to-real settings.

Finally, the writing is also generally not up to the standards of the conference with some grammatical mistakes and awkwardness throughout. The manuscript should be proofread for grammar and style.

[1] Yu, W., Tan, J., Liu, C. K., & Turk, G. (2017). Preparing for the unknown: Learning a universal policy with online system identification. arXiv preprint arXiv:1702.02453.
[2] Lee, J., Hwangbo, J., Wellhausen, L., Koltun, V., & Hutter, M. (2020). Learning quadrupedal locomotion over challenging terrain. Science robotics, 5(47), eabc5986.
[3] Kumar, A., Fu, Z., Pathak, D., & Malik, J. (2021). Rma: Rapid motor adaptation for legged robots. arXiv preprint arXiv:2107.04034.
[4] Ramos, F., Possas, R. C., & Fox, D. (2019). Bayessim: adaptive domain randomization via probabilistic inference for robotics simulators. arXiv preprint arXiv:1906.01728. (in RSS 2019).

**Quality Of The Limitations Section:**

2

**Questions For Rebuttal:**

Please see above, especially the writing of the main method section.

**Robotics Focus:**

4

**Summary Of Paper:**

The paper proposes "domain contraction" with parameter-conditioned policies as a method for learning robust manipulation primitives. The state space is augmented to include latent model parameters (e.g., friction, mass), and the resulting parameter-conditioned value function and policy are derived. At deployment time, given a rough distribution over the latent properties, a corresponding robust policy can be efficiently retrieved. This approach unifies domain randomization and domain adaptation, with both being extreme cases of the proposed domain contraction. Rapid value learning and policy retrieval is enabled by tensor train decomposition and recent work applying it for control.

**Summary Of Recommendation:**

Overall, the paper needs significant re-writing for clarity and proper contextualization.

---

### Official Review · Reviewer_q6Sa · 2024-07-20

**Originality:** 3
**Technical Quality:** 3
**Clarity Of Presentation:** 3
**Potential Impact:** 3
**Recommendation:** 3
**Confidence:** 4

**Review:**

Strengths: Overall, I quite like the idea of efficiently mixing parameter-conditioned policies at runtime, and I think the TT framework is a nice way to achieve this mixing efficiently. The paper is overall well-written and technically solid.

I only have minor concerns:
- It would help to provide some intuition on the scalability of such approaches to higher-dimensional systems or parameter spaces - how high is too high? It is discussed briefly in the conclusion that dimensionality is an issue due to discretization, but some concrete numbers would help to ground this claim.
- Where does the scalability challenge arise? Is this at training time, or at policy retrieval time? Some discussion would help here.
- In Table 2, what is the relation between training time and policy retrieval time?
- For the considered tasks, how important is the effect of parameter discretization density to computation time, and how sensitive is policy performance to a coarse parameter discretization?
- Section 3 would benefit from an overview of the method, before jumping directly into a theorem and proof
- In Eq(1), f is not defined (I suppose it is the dynamics)
- Figure 2 could be clearer - perhaps write explicitly in the caption that the blue and red lines represent explicitly the parameters that each policy is conditioned on.

**Quality Of The Limitations Section:**

3

**Questions For Rebuttal:**

please see review

**Robotics Focus:**

4

**Summary Of Paper:**

In the context of generalization in manipulation planning, the paper proposes domain contraction as a tradeoff between domain adaptation and domain randomization, where a policy is synthesized that is robust to a subset of possible parameters, as opposed to all possible (randomization) or a single parameter (adaptation). To efficiently compute such a policy, the tensor-train framework is used. The method is evaluated on a number of simple manipulation tasks, including on hardware.

**Summary Of Recommendation:**

I recommend weak accept: an interesting concept, clearly-written, and good experimental evaluation.

---

### Official Review · Reviewer_8Guh · 2024-07-21
**The given paper is a good extension of the Tensor Train framework to approximate value functions and learn parameter-conditioned policy through domain contraction. The work has good theoretical insights however, lacks a similar rigor in real-world experiments and no benchmarks.**

**Originality:** 4
**Technical Quality:** 4
**Clarity Of Presentation:** 4
**Potential Impact:** 3
**Recommendation:** 3
**Confidence:** 4

**Review:**

Overall, the authors propose a very interesting framework to unify the domain randomization and domain adaptation methods to learn robust manipulation primitives. However, the manuscript can be difficult to follow since a lot of work is extended from prior work. Given the length restriction, it is understandable that its difficult, but authors can try to include a more detailed preliminary section in the appendix than what they have currently in Appendix Section B. Furthermore, there is no comparison with current well-established methods.

Strengths:
1. The theorems mentioned are well-defined and demonstrate how domain contraction is a unification of domain randomization and domain adaptation.
2. It is a good extension of the TTPI work for learning a parameter augmented value function and the advantage function in TT format.
3. The TT architecture definitely improves the overall efficiency of learning a value function and the corresponding advantage function.


Weaknesses:
1. The reader needs to understand the old literature on tensor train to understand the proposed work. Thus, authors should definitely consider adding more information on TTPI and the Tensor-Train framework relevant to their proposed approach.
2. A lot of the key algorithms and frameworks used are just cited by the authors rather than explaining their relevance in their aforementioned task and learning framework.
3. The paper currently lacks insights into why certain design choices were made. For example, why use TTGO to retrieve the policy?
It is clear that the TT method is better in terms of training time. However, currently, as a researcher/reader, I am not convinced as to why one should use this method over imitation learning or model-based learning that have shown good results even on more complex manipulation tasks.

**Quality Of The Limitations Section:**

2

**Questions For Rebuttal:**

1. The current method needs to be benchmarked with at least one of the well-known methods that learn primitives for a contact-rich manipulation task

**Robotics Focus:**

4

**Summary Of Paper:**

Problem Studied by Authors: The authors of this work focus on the problem of learning robust primitives to execute contact-rich manipulation tasks. The main problem solved by the authors is to generate robust policies that can generalize to a diverse range of tasks. In this work, the authors have focused on 3 of such tasks.  Proposed Solution: Specifically, the authors propose a bi-level approach where they first learn a policy using multiple models and use a tensor train to approximate a value function and then retrieve a policy via domain contraction that extends domain randomization and domain adaptation.

**Summary Of Recommendation:**

Given that there are no benchmarks provided, I can change my weak accept to a strong accept if the authors can provide some sort of benchmarking results and compare their method with relevant primitive learning methods.

---

### Author Rebuttal · Authors · 2024-08-12

We have revised the main paper and added more information to the appendix based on the comments provided by the AC and reviewers. The changes we have made include:
1. Additional information about tensor train and related algorithms in the appendix.
2. Updates to the Related Work section to clarify how our approach relates to existing methods.
3. Revisions to the Method section to enhance readability.
4. Comparisons with two widely used robust policy learning methods.

For your convenience, we have merged the main paper and appendix into a single document in the attachment. The modifications are highlighted in blue font.

---

### Decision · Program_Chairs · 2024-09-04

**Decision:**

Accept

**Comment:**

This paper proposes an interesting extension of the Tensor Train framework, generalizing domain randomization and domain adaptation. The theory is solid, and meaningful evaluation is provided, including on real robot hardware. However, understanding the paper requires reading prerequisite papers first, and core concepts are introduced with insufficient context and prior motivation. Readability here is really a serious issue. In particular, Reviewer aceS gives good advice on how to improve the writing. Technical issues raised by multiple reviewers include scalability to higher-dimensional observations, and (theoretical and experimental) comparison to simpler baselines.

The rebuttal addressed most of the reviewers' concerns, including all mentioned in the preceding paragraph.